# Can Nature-Based Solutions (NBSs) for Stress Recovery in Green Hotels Affect Re-Patronage Intention?

Sunmi Yun [1] and Taeuk Kim [2,*]

1    Department of Hospitality and Tourism Management, Sejong University, Seoul 05006, Korea; sunmiyun80@gmail.com
2    Department of Hotel & Restaurant Management, Kyonggi University, Seoul 03746, Korea
*    Correspondence: teokim1305@naver.com; Tel.: +82-10-4307-3903

**Abstract:** Our research framework in this paper investigated natural-based solutions (NBSs) at green hotels. We employed attention restoration theory (ART) to test the mediating effect of perceived stress (PS), psychological wellness (PW), satisfaction (SA), and the moderating effect of health consciousness (HC) on re-patronage intentions (RI). Data were collected through a survey of 544 customers who frequently visited green hotels in Korea, and structural equation modeling (SEM) was used to test the research hypotheses. The findings generally supported the hypothesized associations of the study variables within our proposed theoretical framework (PS, PW, SF) in order of the mediating effect on RI and confirmed the moderating effect of HC. In addition, the study's results have important theoretical and practical implications for the environment. In the former case, our results demonstrate the application of ART and NBS by explaining the effect of the relationship among PS, PW, and SF on RI and confirm the mediating effect of the ART (PS, PW, SF) on RI, as demonstrated in previous studies. Moreover, in the latter case our results may encourage green hotels to participate in the prevention of environmental problems.

**Keywords:** natural-based solutions (NBSs); attention restoration theory (ART); perceived stress (PS); psychological wellness (PW); mental health (MH); well-being (WB); satisfaction (SA); health consciousness (HC); re-patronage intention (RI); green hotels



## 1. Introduction

Research shows that the official COVID-19 casualty should be double the number that has actually appeared. The actual casualties should be multiplied by $1.5\times$ from the official report for the USA; for Japan only one tenth of casualties have been counted [1]. The Centers for Disease Control and Prevention (CDC) reported a higher lethality rate for COVID-19 during 2021 compared to 2020. This indicates that multiple elements, such as low vaccination rate, low mask usage rate, eased social distance, and the highly contagious Delta variant are in effect [2]. Nations equipped with effective anti-COVID prevention measures are now transitioning to normal life under the "With Corona" program. However, as the Omicron variant appears, we are under an increasing period of COVID-19 patients as the hazard of COVID-19 rises. COVID-19 is damaging the global industry. The tourist industry (i.e., hotel, travel agency, airlines) suffered the most, as COVID-19-related travel attenuation caused USD 1.2 trillion to USD 3.3 trillion in losses; this is close to 2.8–4.2% of global GDP [3]. The global tourist industry took three steps under COVID-19: Challenge → Resilience → Transformation. During the resilience stage, there were positive effects from regional affection. Credits from consumer and related businesses now include the sustainability, well-being of society, climate action, and local communities, which expect to take an imperative role during the transformation phase [4].

A warning about climate change in the World Economic Global Risk Report 2021, which began publishing in 2013, supports this issue. They report that the past two decades

saw natural capital decrease by 40% [5]. The natural capital is being systematically depleting. However, at the same time, nature and its ecosystem services are at the center of the hotel industry business proposition, from food and beverage offers to guests' enjoyment of the natural landscape at the destination. Nonetheless, nature is a "capital" component available to hotels and a natural-based solution (NBSs) to mitigate and adapt to climate change and protect biodiversity while ensuring human mental health (MH) and well-being (WB) [6]. NBS represents an activity to protect, manage, and restore the natural and deformed ecosystem, sustainably providing the WB of humanity, benefits of biodiversity, and adaptive, efficient, and rapid solutions to mend social issues simultaneously [7]. Eco-friendly environments such as WB parks, green cities, gardens, and urban forests can reduce stress, depression, attention deficit hyperactivity disorder, and cardiovascular disease, and improve pregnancy and respiratory symptoms in physiological and psychological ways [8,9]. NBS is expanding to be an important social phenomenon. Thus, an efficient realization of NBSs can improve MH and WB along with other various environmental and social tasks [10,11]. Green hotels can positively influence customer satisfaction (SA) with their eco-friendly structure, spaces, and items. Influences such as reduced stress and depression are a representative sample [12,13].

Attention restoration theory (ART) has been actively studied in environmental psychology due to the benefits NBSs provide to people. In previous studies, people stayed in a natural environment and green space to which it is applied to relieve stress [14,15]. According to that, people follow NBSs to enjoy leisure in nature, reducing their stress and restoring their attention [16,17], which are affected by the stress that comes from dwelling in urban society [18,19]. MH, emotional WB, SA, and stress are the core factors to help explain customers' post-purchase activity. Thus, NBSs and ART take key roles in customer behavior study. Improving customers' MH and WB is now the head issue in the global customer service and tourism business [20,21]. However, understanding the eco-friendly concepts in inducing customers' good MH and WB is not deep enough, even though NBS is taking a pivotal role in the area, as is the study of inducing customers to green hotels. With that said, promising customer maintenance and SA can be a possible output of efforts to effectively maintain customers' MH and WB effectively [21,22]. Previous studies of the tourist industry based on NBSs [13,23,24] have proven that NBSs can enhance human MH and reduce stress though direct and indirect influences. This can be highly related to people's behavior, intention, or attitude.

During the COVID-19 pandemic, travelers collected health knowledge and maintained strict policies and norms to protect their health. To prevent the spreading of COVID-19 among future travelers, health consciousness (HC) will take a vital role in the tourist industry [25]. Research shows that customers with high HC are more likely to avoid or cancel a tour instead of joining health-related activities to make travel [25].

Examining victims of the COVID-19 pandemic from their stress status to perceived wellness will enhance the current study of NBS and ART, and may provide practical implications as well. However, the study of psychological wellness (PW), perceived stress (PS), and re-patronage intention (RI) grafted with NBS and ART is not deep enough currently. This study aims to create a merged model of NBS and ART, sampling guests at South Korea's green hotel to identify the green hotels' structure of PS, PW, SA, and RI. We shed light on the mediating effects of PW, PS, and SA on the relationship between NBSs, green hotels, and RI. At the end of the study, we verify the HC within society under the conditions of the COVID-19 crisis.

## 2. Conceptual Framework

### 2.1. Green Hotels as Natural-Based Solutions

NBSs are not a new concept. It is close to an umbrella concept, including nature-based approaches such as ecosystem-based adaptation or ecosystem-based mitigation [7]. Ultimately, NBS aims to help achieve sustainable development goals (i.e., long-term food security, climate change, water security, human health, disaster management, society, and

economic development). While the definition of NBS is not agreed upon under universal consensus yet, it is known as a helpful approach in various studies.

In a service-related NBS study [26,27], NBS is defined as "a method of improving/enhancing ecosystems to protect nature and solve various social/environmental/economic issues by performing sustainable management and restorations" [28]. From the view of the IUCN and EU, NBSs can be interpreted as a solution to improve natural environments' vulnerable points with social and environmental benefits to maximize the positive effect. The OECD's perception of NBSs emphasizes better healing from COVID-19's affects. They insist on valuing natural resources to secure the health of nature. In their concept, ecosystem services are indispensable for human health; they are essential to healing our society from pandemic damage. Mainly, they are focused on forbidding lumber harvesting and land utilization that can be related to the spread of COVID-19.

NBSs fall into three categories: (1) NBSs related to green infrastructure [26,29]; (2) NBSs related to climate change mitigation and adaptation [30]; and (3) NBSs related to ecosystem services [26,27]. These human health studies related to NBSs share one thing: they are all superficial. Most of the studies focus on the three concepts mentioned above and discuss basic ideas, which indicates that NBS is mainly about environmental and ecosystem issues. Moreover, using the NBS in city planning is essential to improve residents' mental health and welfare and resolve various social tasks [11,31].

Embodying NBS in the hotel industry means improving customer MH and WB. Thus, it has great salience [24]. Green-infra-related studies such as green spaces, green surfaces, green items, and nature play an imperative role in eco-friendly hotels, and structures where the service performance and character can directly lead to the product quality evaluation [32–34]. Human health can be affected by the surrounding social and physical environment; when someone collaborates with nature, nature can improve their MH [24]. Therefore, we can confirm the importance of NBS in human WB. Further studies to determine the benefits of NBSs and their effects should be performed.

## 2.2. Attention Restoration Theory Linked to Natural-Based Solutions Such as Green Hotels

Early studies of stress focused on physical harm and stress applied by accident or by any change [35]. However, recent studies have found that stress could be one reason for various mental sicknesses, including neurosis, depression, anxiety, suicidal tendency, or disruptive mood dysregulation disorder [36]. Half of the world's population dwells in cities. That residential condition causes stress to negatively affect city-dwellers' mental health. For that reason, people seek nature-based leisure activities to reduce stress and restore their attention [15]. Nature can reduce stress for those who contact it, allowing them to recover their brain and mental fatigue [37,38]. There is evidence that nature can offer positive emotion to improve MH or restore attention [15,31,39].

ART draws on nature's attention-recovering potential [40,41]. ART is an environmental psychology theory explicating nature as a restorative environment to heal the fatigue that results from directional attention [40]. RE refers to the environmental trait of attention restoration. When one loses directional attention capability due to the stress of city dwelling, it can be recovered by recognizing the natural environment's absolution effect, which requires no other extra effort [42]. Moreover, restorative experience results from actual restoration experiences such as distress, attention restoration, mental peace, recharging energy, and vigor [41]. The interest in environmental psychology has been present in nature-based activity for a long time. Active research based on ART is commonly conducted, which is expanding to tourism as healing tourism intended for mental healing increases [15,41,42].

Green hotels use NBSs through a nature-based approach (i.e., ecosystem-based adaptation, ecosystem-based relaxation) [12,13] to build a restorative environment, as nature can decrease PS of people and create a restorative environment by imbuing positive effects on PW (i.e., MM, WB) [24,42,43]. In the green-infra-related aspect of NBSs, the physical green environment such as green spaces, green surfaces, and green items may be essential for green hotels [33,34]. Therefore, the physical green environment inside the green hotel

and the natural environment of the exterior are crucial component of NBSs [11–13]. The green physical environment of green hotels can provide stress reduction, body activity enhancement, health inequality relief, emotion improvement, and increased WB and MH. It will bring various positive effects on the health of individuals and society [1–3,31,44]. Previous studies claim that people who reside in the natural environment and green spaces can be distressed [14,15].

Furthermore, research shows PW and PS of NBSs can affect SA [12,13]; Han et al. [13] affected customer satisfaction by increasing MH and WB perception and reducing stress by NBSs in a study of customer maintenance strategy and green hotel-related NBSs. In Han et al. [12], it was shown that NBSs could affect customers' SA and RI by perceiving MH change. The previous studies mentioned above developed the hypothesis of our study.

**Hypothesis 1 (H1).** *Green hotels as NBSs have a significant effect on psychological wellness.*

**Hypothesis 2 (H2).** *Green hotels as NBSs have a significant effect on perceived stress.*

**Hypothesis 3 (H3).** *Perceived stress has a significant effect on psychological wellness.*

**Hypothesis 4 (H4).** *Perceived stress has a significant effect on satisfaction.*

*2.3. Re-Patronage Intention*

Intentions are subjective judgments against individual activity, and re-patronage intention (RI) usually functions as an outcome variable in the service business. RI can be defined as "An individual judgment to use the assigned service of the same place after considering one's current and possible situation" [45]. If a customer receives RI from a place, one is likely to re-patronize and recommend the place to their acquaintances [46]. This concept assumes that a customer's choice interrelates to loyalty based on user experience [47]. In Butcher [48], RI is claimed to be a measurable service result. On the other hand, Soderlund and Ohman [49] took RI as a decisive intention based on expectation. The study of RI was key to understanding travelers' behavior intentions. Moreover, it evolved as the re-patronage intention derived from the sum of the travelers' satisfaction, behavior intention, and evaluation of travel destination. Naturally, further research into recommending and sharing behavior of favored travel destination became the linchpin [50,51]. Below are related previous studies: Lin [52] showed that customers' mental well-being can affect RI on cruise travel; Kim et al. [21] showed that the well-being of the lounge experience can affect satisfaction and RI; Han et al. [12] examined customers' SA affecting the effect of NBS by increasing MH and WB perception and reducing stress. Previous studies [11,12,21,24,52] developed the hypothesis of our study.

**Hypothesis 5 (H5).** *Psychological wellness has a significant effect on re-patronage intention.*

**Hypothesis 6 (H6).** *Satisfaction has a significant effect on re-patronage intention.*

*2.4. The Mediating Effect of Perceived Stress, Psychological Wellness, and Satisfaction*

Previous studies related to stress reduction [12,13,53,54] show that green spaces, structures, and NBS have a high correlation with customer RI by improving their MH and WB. In a study of green hotel visitors, Han et al. [13] found a significant mediating effect of customer WB and self-rated MH on the relationship between NBSs and customer behavior. This shows that a nature-based restorative environment can reduce the stress level of its users [14]. Moreover, Grahn and Stigsdotter [55] sampled eight environments to prove that restorative environment has the highest stress-reducing potential.

The detailed analysis between average nature-based activity period and emotional WB mediated by the restoration experience and visit shows that restoration experience can affect the WB. This is due to exposure to natural environments and mediating the emotional WB under specific nature-based activities [15]. In Qiu et al. [56], it was shown that the

natural environment could directly affect tourists' attention restoration and quality of life. The sampled Chinese mountaineers and Australian coast visitors showed that, based under ART, examining ART can apply to various natural environments. Moreover, ART can be a valuable tool to improve people's mental health post COVID-19.

**Hypothesis 7 (H7).** *Perceived stress, psychological wellness, and satisfaction will be parallel mediation on the relationship between green hotels as NBS and re-patronage intention.*

*2.5. Health Consciousness as Moderator*

People with HC are highly interested in private health. They are more conscious about health and are likely to improve and join health activities to remain healthy [57]. People with HC are active in doing health-related research. They will act quickly on the information they collect, performing more than one related activity to secure a healthy life [58]. HC can be treated as a psychological state, predicting possible outcomes from health attitude and activity [59]. This can define HC as a preparatory stage to perform a health activity, making people join the health improvement and maintenance activities [60]. Many theories claim that HS can promote disease prevention, and is the first health improvement action [61].

In this study, we will examine HC as a prevention and protection against COVID-19 derived infection and wariness. Each individual reacts differently to recognized health hazards depending on their level of HC [62]. This can lead to a gap in HC depending on their response to health-related messages [63]. People with high HC take the COVID-19 situation as a high health risk and strengthen their health attitude and norms when traveling [64]. Additionally, tourists with high HC are very likely to join the health-related activity under COVID-19 pandemic. This leads them to cancel or avoid their travel plans [25].

The following articles are previous studies of HC and the tourist industry. In Zhang et al. [65], tourists from rural, eco-friendly areas were sampled, showing that travelers with high HC have a high desire to travel for their health and to protect it. In Wu et al. [64], the authors verified COVID-19 awareness and the regulating effect of HC on social distancing among the sampled Chinese hotel employees. This study set the following assumptions based on the previously mentioned studies:

**Hypothesis 8a (H8a).** *Health consciousness plays a significant moderating role in the relationship between green hotels as NBSs and perceived stress.*

**Hypothesis 8b (H8b).** *Health consciousness plays a significant moderating role in the relationship between perceived stress and psychological wellness.*

**Hypothesis 8c (H8c).** *Health consciousness plays a significant moderating role in the relationship between perceived stress and satisfaction.*

**Hypothesis 8d (H8d).** *Health consciousness plays a significant moderating role in the relationship between green hotels as NBSs and psychological wellness.*

**Hypothesis 8e (H8e).** *Health consciousness plays a significant moderating role in the relationship between psychological wellness and satisfaction.*

**Hypothesis 8f (H8f).** *Health consciousness plays a significant moderating role in the relationship between psychological wellness and re-patronage intention.*

**Hypothesis 8g (H8g).** *Health consciousness plays a significant moderating role in the relationship between satisfaction and re-patronage intention.*

## 3. Materials and Methods

### 3.1. Measures and Questionnaire Development

This study focused on (1) expected re-patronage intention of green hotel customers, (2) adding ART to base NBSs though the green hotel model, (3) the utility of NBSs as verified through previous studies of environmental science [7,11,26,30,44,66] and ART through environmental psychology [38,40,44,56,67–69]. With these, we intended to accumulate studies of hotel industry hypotheses in the previous studies mentioned above with a research model based on preset hypothesis.

Measurement variables of previous studies examined the change of customer intentions against green hotels' NBSs, mediated by PS, PW and SA. However, there were insufficient studies related to the hotel industry [12], as most of the variables were from studies of the tourist industry [13,15,23–25,42]. Thus, we developed a measurement variable based on modified previous study material and in-depth interviews made by the hotel manager and CEO.

In detail, we first selected five hotel managers with over ten years of experience and five professors of hotel management who were willing to join the in-depth interview. Second, we made detailed explanations of the purpose of the study and interviews. Third, to check the suitability of the questionnaire, the questionnaire used in the previous study was shown to the experts through a one-on-one interview, and it was verified whether the questionnaire was suitable for green hotels research through the interview. Fourth, by combining five interviews with experts, it was modified and supplemented to fit the green hotels study. Lastly, we deployed the survey to be distributed to twenty customers who had visited the Grand Walkerhill Seoul Hotel in the time span from 11 November 2021 to 13 November 2021 as a pretest under final review, providing transparency to the survey. The Grand Walkerhill Seoul Hotel (independent hotel, 799 rooms) surveyed is green hotel which is located overlooking the Han River and the slopes of the Acha Mountain; the plants and flowers inside the hotel are all natural, and the hotel has large glass windows for natural lighting.

Measurement items shown below (Appendix A) include (1) eight items extracted from [12,13] for "green hotel as NBS"; in this study, an NBS is defined as "a special concept or method; developed to induce social and environmental outputs from COVID-19 and provide a solution for various social and environmental challenges by utilizing nature". Two factors were derived, indoor green hotels (IGH) and outdoor green hotels (OGH), and (2) nine items are extracted from [12,13,70] for PW.

In this study, PW is "an individual's self-assessment of their ability to deal with a mental crisis in order to return to a pre-crisis state." Items stem from two factors (MH, WB) by following the previous studies. (3) Four items were extracted (i.e., I fear I cannot take control of the important matters) from [71] for PS. This study defined PS as "a subjective consciousness against environmental stimulants, variable to individual traits". (4) four items are extracted (i.e., 'as a whole, I have really enjoyed myself at this hotel as expected') from [21,72] for SA. Customers had to NBSs after visiting the green hotel. (5) Four items were extracted (i.e., I have a strong intention to visit the service provider again) from [73] for RI, which is the most reliable factor of behavior intention against the green hotel. This study used 34 items to evaluate six variables, using a Likert five-point scale: 1. Not important at all, 2. Not important, 3. Normal, 4. Important, 5. Very important.

### 3.2. Data Collection Process and Data Analysis

Targets were adults aged 20 and older who visited green hotels within the year. Before the primary survey, we performed a pilot test to find any structural issues with the paper. From 8 November 2021 to 4 November 2021, fifty students under the doctor's hotel management course were applied. We used the online research company embrain.com between 15 November 2021 and 28 November 2021, and received 600 surveys. Fifth-six undependable responses were removed, leaving 544 (90.7%) valid samples for data analysis. The tools for the analysis were SPSS 20.0 for exploratory factor analysis and AMOS 24.0 for

confirmatory factor analysis (CFA), structural equation modeling (SEM), mediator effect, and moderator effect.

## 4. Results

### 4.1. Characteristics of Respondent

Of the 544 respondents, 353 respondents (64.9%) were female and 191 respondents (35.1%) were male. Present marital status was highest for married people (307 respondents, 56.4%), followed by single people (232 respondents, 42.6%), and other (five respondents, 0.9%). Among the participants, 37.7% were 30–39 years old (205 respondents), 11.9% were 20–29 years old (65 respondents), 9.0 were 40–49 years old (49 respondents), and 3.7% were 50 years old or over (20 respondents). The education level was generally high; 328 respondents (60.3%) were college and university graduates, and 146 respondents (26.8%) were in graduate school or higher. Additionally, 70 respondents (12.9%) had attended high school and below. The annual household income of USD 20,000–29,999 (238 respondents, 43.8%) was the highest, followed by USD 40,000 and more (109 respondents, 20.0%), USD 30,000–39.999 (99 respondents, 18.2%), and under USD 20,000 (98 respondents, 18.0%). By job, employed (219 respondents, 40.3%) showed the highest, followed by professionals (113 respondents, 20.8%), housewives (80 respondents, 14.7%), self-employed (74 respondents, 13.6%), and students (43 respondents, 7.9%). The number of hotel visits for one year was as high as 46.5% (253 respondents) for 3–5, followed by 1–2 times 24.6% (134 respondents), 6–9 times 20.2% (110 respondents), and ten and more times 8.6% (47 respondents). The purpose of visiting the green hotel was leisure 69.5% (378 respondents), business 16.9% (92 respondents), and other 13.6% (74 respondents).

### 4.2. Reliability and Validity Assessments and Confirmatory Factor Analysis

The measurement unit was developed from previous studies, to offer content validity. To analyze the measurements' reliability, we performed exploratory factor analysis; If Cronbach's $\alpha$ value is above 0.6, we can say it has internal consistency. During the analysis of all measurement items (NBSs, PW, SA, RI, and HC), one of the PW items ("I feel healthy and happy when staying at this hotel") was removed, as it had a Cronbach's $\alpha$ value below 0.5. Reverse-coding was performed to PS for analysis.

To verify the construct reliability and convergent validity, we performed confirmatory factor analysis (CFA) on the constructs (NBSs, PS, PW, SA, RI, and HC). The IPS helped to increase the scale's reliability when performing the CFA. Parceling is a procedure for computing average scores or sums through multiple items. The average scores or sum instead of the individual item scores serve as indicators of latent factors in the SEM analysis. In this study, items with high correlation or similar factor loadings in factor analysis were grouped using Cattle's radial parceling [74].

We measured IPS based on factor loading for study, derived from exploratory factor analysis of NBSs items (IGH, OGH) and PW items (MH, WB). When the average variance extracted (AVE) and construct reliability (CR) are above 0.5 and 0.7, we can say there is convergent validity [75]. We found an AVE between 0.589 and 0.861 and CR between 0.749 and 0.927; these values satisfy the convergent validity, as displayed in Table 1.

Normally, the goodness-of-fit index should have CFI, IFI, and TLI above 0.9 and below 1.0 [76]. RMSEA should remain below 0.05, and is acceptable between 0.05~0.1 [77]. Our study showed a well-established model to fit the data satisfactorily (goodness-of-fit statistics for measurement models: $\chi 2 = 289.844$, df = 137, $p < 0.000$, $\chi 2/df = 2.116$, RMSEA = 0.092, CFI = 0.929, IFI = 0.931, TLI = 0.902). We present correlation to verify discriminant validity.

After the analysis, all variables showed a square price below AVE value and all correlation coefficients did not include 1 in standard error estimation. Thus, all variables obtained discriminant validity. To verify the discriminant validity, we performed Pearson Correlation Analysis. All variables (NBSs, PS, PW, SA, RI, and HC) showed a significance level of $p < 0.001$, index of the correlation coefficients.

**Table 1.** The measurement model and correction.

| Construct and Scale Item | | Standardized Loading | MEAN (SD) | AVE (CR) | NBSs | PW | PS | SA | RI | HC |
|---|---|---|---|---|---|---|---|---|---|---|
| NBSs | IGH | 0.836 | 4.323 | 0.601 | 1 | | | | | |
| | OGH | 0.709 | (0.567) | (0.749) | | | | | | |
| PW | MH | 0.961 | 4.259 | 0.861 | 0.758 *** | 1 | | | | |
| | WB | 0.893 | (0.749) | (0.925) | (0.575) | | | | | |
| PS | PS1 | 0.737 | | | | | | | | |
| | PS2 | 0.841 | 4.393 | 0.589 | 0.636 *** | 0.597 *** | 1 | | | |
| | PS3 | 0.857 | (0.581) | (0.849) | (0.404) | (0.356) | | | | |
| | PS4 | 0.609 | | | | | | | | |
| SA | SA1 | 0.883 | | | | | | | | |
| | SA2 | 0.940 | 4.487 | 0.848 | 0.730 *** | 0.827 *** | 0.615 *** | 1 | | |
| | SA3 | 0.939 | (0.622) | (0.944) | (0.533) | (0.684) | (0.378) | | | |
| RI | RI1 | 0.837 | | | | | | | | |
| | RI2 | 0.863 | 4.227 | 0.650 | 0.650 *** | 0.857 *** | 0.704 *** | 0.842 *** | 1 | |
| | RI3 | 0.901 | (0.641) | (0.878) | (0.423) | (0.734) | (0.496) | (0.709) | | |
| | RI4 | 0.681 | | | | | | | | |
| HC | HC1 | 0.878 | | | | | | | | |
| | HC2 | 0.781 | 3.882 | 0.762 | 0.404 *** | 0.486 *** | 0.404 *** | 0.567 *** | 0.557 *** | 1 |
| | HC3 | 0.952 | (0.951) | (0.927) | (0.163) | (0.236) | (0.163) | (0.321) | (0.310) | |
| | HC4 | 0.873 | | | | | | | | |

Note 1. NBSs = natural-based solutions, IGH = indoor of green hotels, OGH = outdoor of green hotels, PS = perceived stress, PW = psychological wellness, MH = mental health, WB = well-being, SA = satisfaction, RI = repatronage intention, HC = health consciousness. Note 2. Goodness-of-fit statistics for the measurement model: $\chi2 = 289.844$, df = 137, $p < 0.000$, $\chi2/df = 2.116$, RMSEA = 0.092, CFI = 0.929, IFI = 0.931, TLI = 0.902). Note 3. All factor loadings are significant at $p < 0.001$ (***). Correlations between variables are below the diagonal. Squared correlations between variables are within parentheses

### 4.3. Research Hypotheses Testing and Structural Equation Modeling

Variables of the research model were the following items: Inner variable, NBSs; extraneous variable, RI; intervening variables, PW, PS, and SA. We used a covariance matrix and maximum likelihood estimation for the analysis. Goodness-of-fit statistics were favorable ($\chi2 = 141.909$, df = 77, $p < 0.001$, $\chi2/df = 1.843$, RMSEA = 0.080, CFI = 0.960, IFI = 0.961, TLI = 0.938). Additionally, SEM showed high prediction power for RI n general ($R^2 = 0.819$). The standardized path coefficients and t-values are shown in Table 2. The hypothesis test results are provided in Figure 1. The path estimates show that NBSs had a positive direct effect on ascription of PW ($\beta = 0.648$, $p < 0.001$) and PS ($\beta = 0.541$, $p < 0.001$); thus, H1 and H2 were supported. The result of estimation indicated that PS had a significant positive effect on PW ($\beta = 0.327$, $p < 0.01$) and PS ($\beta = 0.291$, $p < 0.01$); thus, H3 and H4 were supported. As expected, PW had an impact on SA ($\beta = 0.639$, $p < 0.001$) and RI ($\beta = 0.427$, $p < 0.001$); H5 and H6 were supported. It was found that SA had a significant positive effect on RI ($\beta = 0.292$, $p < 0.05$); thus, H7 was supported.

We found direct and indirect effects of NBS on RI related to PW, PS, and the mediating effect of SA. To verify the significance of indirect effect, we used the parametric bootstrapping, and the result is displayed in Table 2. The findings revealed that NBSs significance affected PCBI ($\beta$ NBSs → PW and PS and SA → RI = 0.692, $p < 0.001$) indirectly through PW, PS, and SA, thus confirming them as partial mediating variables. Therefore, H8 was supported.

**Table 2.** Structural model results and hypotheses testing.

| Hypothesized Paths | Coefficients | *t*-Values |
|---|---|---|
| H1: NBSs → PW | 0.648 | 5.607 *** |
| H2: NBSs → PS | 0.541 | 4.156 *** |
| H3: PS → PW | 0.327 | 2.842 ** |
| H4: PS → SA | 0.291 | 3.098 ** |
| H5: PW → SA | 0.639 | 6.921 *** |
| H6: PW → RI | 0.427 | 3.960 *** |
| H7: SA → RI | 0.292 | 2.467 * |
| Explained variable: | $R^2$(PW) = 0.629 | $R^2$(PS) = 0.420 |
| | $R^2$(SA) = 0.745 | $R^2$(RI) = 0.819 |
| Indirect effect: | β (NBSs → PW& PS & SA → RI) = 0.692 *** | |
| Total effect on RI: | β(RI) = 0.692 *** | |
| The results: | H8 supported | |

Note 1. NBSs = natural-based solutions, PS = perceived stress, PW = psychological wellness, MH = mental health, WB = well-being, SA = satisfaction, RI = re-patronage intention. Note 2. Goodness-of-fit statistics for the structural model: $\chi2$ = 141.909, df = 77, $p < 0.001$, $\chi2$/df = 1.843, RMSEA = 0.080, CFI = 0.960, IFI = 0.961, TLI = 0.938. Note 3. * $p < 0.5$, ** $p < 0.01$, *** $p < 0.001$.

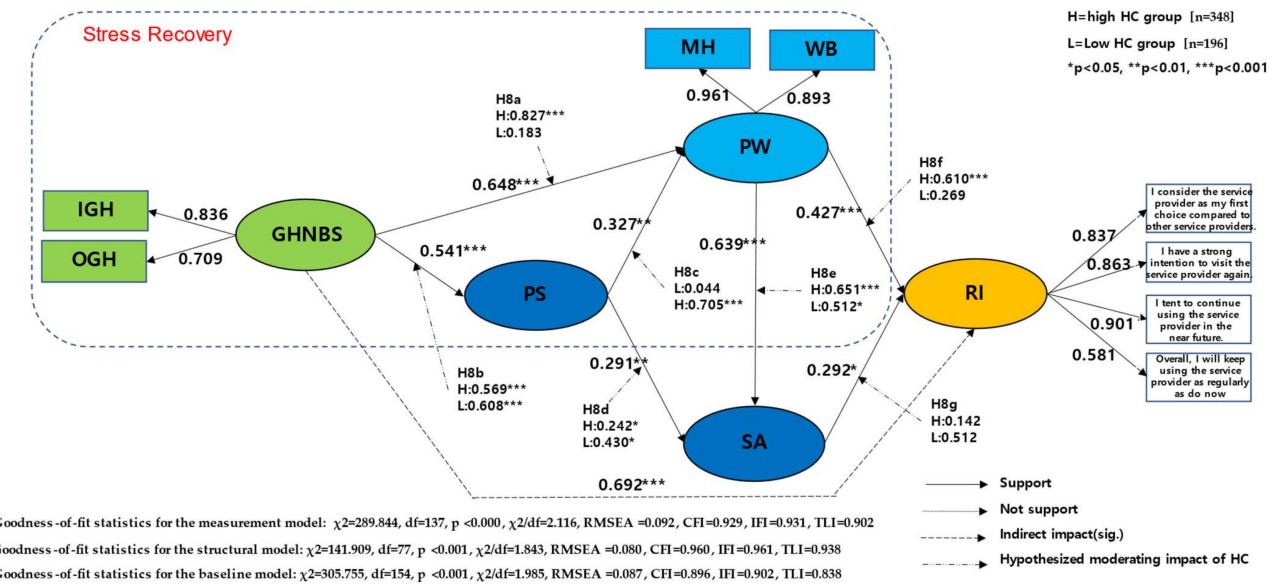

**Figure 1.** Structural equation model estimation and test for structural metric invariance.

### 4.4. Moderating Effect of Health Consciousness

We performed K-mean cluster analysis on moderating variable HC to check whether NBSs build a significant HC gap on RI. The result was in two groups, one with high HC ($n = 322$) and one with low HC ($n = 207$). The baseline model showed an acceptable level for the data suitability (goodness-of-fit statistics for the baseline model: $\chi2 = 305.755$, df = 154, $p < 0.001$, $\chi2$/df = 1.985, RMSEA = 0.087, CFI = 0.896, IFI = 0.902, TLI = 0.838).

If the constrained model and unconstrained model's chi-square value gap exceed 3.84 and reach a significance level ($p < 0.5$), we can say there is a regulation effect (as shown in Table 3 and Figure 1). The chi-square test results with measurement invariance revealed that GHNBS-PW path ($\Delta\chi2$ [1] = 11.398, $p > 0.05$) between high and low HC groups and PS-PW path ($\Delta\chi2$ [1] = 1O.624, $p > 0.05$) between high and low HC groups did differ significantly, and more than 3.83 for chi-square difference. Hence, this result supports Hypothesis H9a and H9c. However, H9b, H9d, H9e, and H9F could not be supported, as they did not show significant results.

**Table 3.** Result of moderating effect.

| Paths | High-HC Group ($n$ = 348) | | Low-HC Group ($n$ = 196) | | Nested Model Constrained to Be Equal |
|---|---|---|---|---|---|
| | Coefficients | $t$-Values | Coefficients | $t$-Values | |
| H8a: NBSs-PW | 0.827 | 3.394 *** | 0.183 | 1.266 | $\chi2$ (155) = 317.15 3 [a] |
| H8b: NBSs-PS | 0.569 | 3.936 *** | 0.608 | 3.642 *** | $\chi2$ (155) = 305.94 1 [b] |
| H8c: PS-PW | −0.044 | −0.280 | 0.705 | 4.267 *** | $\chi2$ (155) = 316.37 9 [c] |
| H8d: PS-SA | 0.242 | 2.179 * | 0.430 | 2.071 * | $\chi2$ (155) = 306.18 5 [d] |
| H8e: PW-SA | 0.651 | 5.793 *** | 0.512 | 2.576 * | $\chi2$ (155) = 305.882 [e] |
| H8f: PW-RI | 0.610 | 4.626 *** | 0.269 | 1.179 | $\chi2$ (155) = 306.60 2 [f] |
| H8g: SA-RI | 0.142 | 1.094 | 0.512 | 1.742 | $\chi2$ (155) = 306.41 2 [g] |

| | | |
|---|---|---|
| Chi-square difference test:<br>[a] $\Delta\chi2$ (1) = 11.398, $p$ > 0.05<br>[b] $\Delta\chi2$ (1) = 0.186, $p$ > 0.05<br>[c] $\Delta\chi2$ (1) = 10.624, $p$ > 0.05<br>[d] $\Delta\chi2$ (1) = 0.430, $p$ > 0.05<br>[e] $\Delta\chi2$ (1) = 0.127, $p$ > 0.05<br>[f] $\Delta\chi2$ (1) = 0.847, $p$ > 0.05<br>[g] $\Delta\chi2$ (1) = 0.657, $p$ > 0.05 | Test results:<br>H8a: Supported<br>H8b: Not support<br>H8c: Supported<br>H8d: Not supported<br>H8e: Not support<br>H8f: Not supported<br>H8g: Not supported | Goodness-of-fit statistics for the baseline model:<br>$\chi2$ = 305.755, df = 154, $p$ < 0.001, $\chi2/df$ = 1.985,<br>RMSEA = 0.087, CFI = 0.896, IFI = 0.902, TLI = 0.838 |

Note 1. NBSs = natural-based solutions, PS = perceived stress, PW = psychological wellness, MH = mental health, WB = well-being, SA = satisfaction, RI = re-patronage intention, HC = health consciousness. Note 2 * $p$ < 0.5, *** $p$ < 0.001.

## 5. Results and Discussion

### 5.1. Conclusions

As the COVID-19 pandemic increases interest in social and environmental issues, the importance of NBSs has come to the forefront [13,78]. The COVID-19 pandemic can cause stress, while green hotels can reduce stress; thus, we performed this study to find green hotel NBSs' customer mental care ability and effect on SA and RI by restoring PS.

This study had three goals. First, to merge NBS and ART to investigate a model of eco-friendly hotel and PS, PW (MH, WB), SA, and RI. This study can provide standard data for a solid practical implication, exceeding the limits of previous studies. Second, we shaped the verification results of previous studies by examining the multiple mediator effects of PW, PS, and RI drawn from green hotel NBSs and RI. Third and last, we tried to shed light on the mediating role of HC against NBSs, PW, PS, SA, and RI. After COVID-19 occurred, people showed behavioral and perceptual changes. Customers who had COVID-19 developed relatively high HC compared to those who had not. Thus, monitoring the behavior change was considered vital to us.

Our empirical analysis shows that NBSs can significantly impact PW and PS. This result supports previous research [37,70]. The significant impact of PS on PW and SA matches the results of previous studies [79]. Additionally, the impact of PW on SA and RI matches previous studies [12,13]. The partial mediation effect of PS, PW, and SA between NBSs and RI matches previous studies as well [79]. Moreover, we found a mediating effect of HC on NBSs, PW, PS, SA, and RI. This supports H9a and H9b and partially supports previous studies [64,65].

### 5.2. Theoretical Implication

The research implications of this study identified the positive effect of green hotel NBS. We applied ART to supplement PS, expanding the academic field of NBSs to stress restoration. In the past, NBSs focused on recognizing improvement of human MH and WB. This study applied ART to NBSs, re-examining NBSs' stress restoration aspect of reducing PW and PS. It is relevant that we managed to examine what NBSs does to PS, SA, and RI via applying ART. It was relevant to examine what restorative experience can do for customer satisfaction and RI.

After the COVID-19 pandemic, people recognized health risks more than ever which raised HC. Additionally, people with high HC are likely to make healthier lifestyle decisions. Thus, there is a theoretical implication around examining HC as a moderator between PW, PS, SA, and RI, as this was the first empirical research about the topic.

### 5.3. Practical Implication

The practical implications of this study include first, that green hotels' NBSs can help PW (MH, WB) and reduce PS. From this, green hotels' NBSs can reduce stress caused by COVID-19. Moreover, stress is the core factor of various diseases. Thus, hotel managers should secure an eco-friendly environment and nature accessibility for customers looking for the stress-reducing effect of a green hotel. Deploying roads and signs to mountains, rivers, and oceans near green hotels can increase nature accessibility. Additionally, installing an eco-friendly bench will offer a fine rest in nature.

Second, customers with perceived stress visiting green hotels affect psychological wellness (mental health, well-being) and satisfaction, although not as high. In order to alleviate perceived stress, healing through a rehabilitation program using nature may be helpful. In addition to the healing and recovery environment provided by nature, stress can be managed through leisure activities that can be carried out in nature. For example, in a green hotel in nature, stress management can be performed by healing the tired physical conditions and mind by creating programs such as healing yoga in the forest.

Third, customers with high SA visiting green hotels had PW which was high by the characteristics of NBSs, and low on RI. As a result of this study, it can be seen that the variable of customer's SA at green hotels does not have significant effect on RI. On the other hand, green hotel customers seem to be satisfied through MH and WB from rest. However, the purpose of hotel customers' visits (restaurant visit, casino, leisure (swimming pool, tennis court, kids' activity, etc.) should be considered in various ways. It is important to ensure that these things and the advantages of the Green Hotel can be accompanied at the same time as RI. Specifically, green hotels should supplement suitable facilities so that customers can feel attracted to the green physical environment. For example, green hotels should make a hotel swimming pool close to nature for customers who feel like swimming in a river or sea and provide the comfort of an indoor swimming pool.

Fourth, while NBSs of green hotels did not directly affect RI, it was found to have a significant effect on PW, PS, and SA as indirect effect. These results show that the characteristics of green hotels' NBSs help customers recover from stress, increasing their willingness to revisit intentions. By increasing the accessibility of green hotels, the NBS characteristics of green hotels can have a positive effect on customers' stress recovery. For example, in order to compensate for the geographical shortcomings of green hotels that must be close to nature, transportation that makes them easily accessible to green hotels located outside should be provided. Such a pickup service (i.e., limousine or hotel shuttle bus) is provided for convenient access from airports or train stations located around green hotels.

Fifth and last, the moderation effect of HC was valid only on NBSs and PW and PS and PW. It was ineffective on HC and other combinations between NBSs and PW, PS, and SA on RI. Furthermore, if the HC is not high enough, it is not practical for any group. COVID-19 incursion raised the HC of people generally. Additionally, high HC can affect the relationship between NBSs and PW. We found that a group with high HC on the relationship between PS and PW (MH, WB) has significant effect. Plus, a previous study by [25] defined HC as a critical factor in preventing COVID-19 contagion. A group with high HC is likely to cancel or change their travel plans when the destination is under high infection risk. Thus, it seems that anxiety towards society is the catalyst that boosts the effect of green hotels' NBSs and PS on PW. Nevertheless, as the COVID-19 situation is not yet resolved, a secure quarantine system should be considered to reduce customers' anxiety.

### 5.4. Limitations

This study has several limitations. First, the research method was limited to cross-section; measurements were made only from recognizing a limited period and aspect. A longitudinal study of change over time should be carried out, i.e., comparing pre-COVID and post-COVID data to check the effect of green hotel NBS on PW, PS, SA, and RI in order to collect diverse data. Second, the target was limited. Continuing research may target the

food service industry (i.e., eco-friendly restaurants or coffee houses) to find more valuable factors. Third and last, this study focused on the green hotel visiting experience only. Continuing research may differentiate the sample and examination by target area (i.e., rural and urban areas) to find any significant gaps between them. Lastly, although this study was conducted on customers who visited green hotels located in Korea, future research needs to expand on this and compare customers from Asia, Europe, and the Americas in addition to Korea.

**Author Contributions:** Conceptualization, S.Y. and T.K.; methodology, S.Y.; formal analysis, S.Y.; investigation, T.K.; resources, T.K.; data curation, T.K.; writing—original draft preparation, S.Y. and T.K.; writing—review and editing, S.Y. and T.K. All authors have read and agreed to the published version of the manuscript.

**Funding:** This research received no external funding.

**Conflicts of Interest:** Authors declare no conflict of interest.

## Appendix A. Measurement Items

| **Indoor of the green hotel [12,13]** |
| --- |
| I easily see green interior decorations and diverse living plants in the lobby area of this hotel. |
| I easily see a variety of green items and light through glass windows in this hotel's restaurants. |
| I easily see diverse flowers, trees, and potted plants in the hotel's coffee lounge. |
| green space can easily be seen everywhere in this hotel. |
| **Outdoor of the green hotel [12,13]** |
| This hotel has easy access to the natural environment (i.e., mountains, forests, rivers, seas, lakes, natural parks). |
| The region surrounding this hotel has good weather (i.e., temperature, humidity, and precipitation). |
| The region surrounding this hotel has good and fresh quality air. |
| The region around this hotel is safe from natural disasters (i.e., earthquakes, typhoons, tsunamis, floods). |
| **Mental health [12,13]** |
| Staying at this hotel plays an important role in relieving my mental stress and anxiety. |
| Staying at this hotel helps me rising my confidence in everyday life. |
| Staying at this hotel is worthwhile as it makes me think that I am a valuable and important person. |
| Staying at this hotel is worthwhile as it helps me turn any anxiety and worry into confidence |
| **Well-being [12,13]** |
| I feel healthy and happy during my stay in this hotel. |
| I feel emotionally secure during my stay at this hotel. |
| This hotel plays an important role in making my mind calm and peaceful. |
| Thanks to this hotel, I was able to relax comfortably. |
| Thanks to this hotel, I was able to refresh my mood. |
| **Perceived stress [35]** |
| For the past 30 days, I have been unable to control the important things in my life. |
| For the past 30 days, I have not been able to handle my personal problems. |
| In the past 30 days, my work has not gone my way. |
| In the past 30 days, I have felt difficulties that I cannot overcome. |
| **Satisfaction [72]** |
| Overall, I am satisfied with my experience at this hotel. |
| My decision to stay at this hotel was a wise one. |
| As a whole, I have really enjoyed myself at this hotel as expected. |
| **Health consciousness [64]** |
| I often reflect on my health a lot and try to protect it. |
| I'm awfully self-conscious about my health. |
| I am generally watchful to my inner feelings about my health. |
| I am incessantly checking my health. |
| **Re-patronage intention [73]** |
| I consider the nature-based solution of this hotel is better compared to other hotels. |
| I have a strong intention to visit this hotel again. |
| I intend to continue using this hotel in the near future |
| Overall, I will keep using this hotel as regularly as I do now. |

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
