# Peer review of "Can Nature-Based Solutions (NBSs) for Stress Recovery in Green Hotels Affect Re-Patronage Intention?"

_sustainability, doi:10.3390/su14063670_

Round 1
Reviewer 1 Report
Regarding perceived stress (PS), should it be about the employees in hotel or visitors staying in the hotel? Readers are very confused about that. Literature review (lines 122-129) talked about employees. However, the remaining of the paper seems to talk about visitors. Why is there stress among visitors during hotel stay?
A more comprehensive research design is needed for the paper. How were the in-depth interviews with hotel management linked with the surveys on hotel visitors?
How were the interviewers and survey respondents sampled? Did the sampling method result in sample selection bias? If yes, this should be mentioned in the limitations section. Also, the authors should explain how the bias affected the interpretation of the research findings.
There are too many hypotheses in the paper. The discussion of the analysis results is not adequate at all. Too much emphasis has been placed on the statistical part.
The paper has a lot of typos and grammatical errors. E.g. Should it be “and” rather than “an” in line 262? What does “c” in “over ten years c of experience” (line 264) stands for? The authors should have the paper proofread by a professional English writer before submission.
Author Response
Response to Reviewer 1 Comments
Manuscript ID: sustainability-1614004
Title: Can natural-based solutions (NBSs) for stress recovery in green hotels affect re-patronage intention?
Dear Editor and reviewers,
Thank you very much for reviewing our manuscript. We appreciate the opportunity you have afforded us to revise and resubmit. We found your suggestions to be thought-provoking and useful and have worked diligently to improve the manuscript as you suggested. Below, you will find our replies and responses to your constructive comments. Within the responses, red sections denote changes we made to the manuscript itself. Within the manuscript itself, changes are highlighted in red. We hope the changes made are satisfactory to you.
******************************************************************************************
Point 1: Regarding perceived stress (PS), should it be about the employees in hotel or visitors staying in the hotel? Readers are very confused about that. Literature review (lines 122-129) talked about employees. However, the remaining of the paper seems to talk about visitors. Why is there stress among stay?
Response 1: Thank you for the helpful comments. We have corrected some confusing parts in the text. This study was conducted with customers of green hotels.
Point 2: A more comprehensive research design is needed for the paper. How were the in-depth interviews with hotel management linked with the surveys on hotel visitors? How were the interviewers and survey respondents sampled? Did the sampling method result in sample selection bias? If yes, this should be mentioned in the limitations section. Also, the authors should explain how the bias affected the interpretation of the research findings
Response 2: Thank you for your comment. We have been rewritten "3.1. Measures and Questionnaire Development" by reflecting the opinions of the reviewer. Please see line 260-270, in page 6.
Point 3: There are too many hypotheses in the paper. The discussion of the analysis results is not adequate at all. Too much emphasis has been placed on the statistical part.
Response 3: Thank you for your comment. From our point of view, these are necessary hypotheses, so I couldn't modify them. However, the discussion of the analysis results was revised in general. Please see line 431-457, in page 11-12.
Point 4: The paper has a lot of typos and grammatical errors. E.g., Should it be “and” rather than “an” in line 262? What does “c” in “over ten years c of experience” (line 264) stands for? The authors should have the paper proofread by a professional English writer before submission.
Response 4: Thank you for your comment. We have corrected the part pointed out by the reviewer correctly.
Reviewer 2 Report
The subject of study, the variables analyzed, in depth, and the approach to the research seems to be of interest to readers of ‘Sustainability’. Natural Based Solutions and Attention Restoration Theory applied to hotel business, green hotel. Mediating effect of Perceived Stress, Psychological Wellness and Satisfaction on Repatronage Intention, and moderating effect of Health Consciousness. The results and conclusions of the research and its practical implications add value to hoteliers and scientific community.
Unfortunately, this paper has several weaknesses that should be addressed to make it publishable:
ABSTRACT: Line 12, satisfaction (SF). Then Line 23, satisfaction (SA). Should it be SA in Line 12?
INTRODUCTION: Satisfaction is briefly mentioned in the Introduction. Line 77 (no abbreviation) and Line 94 (abbreviated). Should this dimension be clarified?
CONCEPTUAL FRAMEWORK: some layout and errors, line 213.
MATERIALS AND METHODS: Line 270 and 271, November, which year? Line 293, dates are not clear (“November 8 to November 4”?). Lines 294-295, November, year?
It might help for the readers to have some information about the “green hotel”, Seoul Walkerhil Hotel. Is this hotel the subject of the main research? Is it privately owned? part of hotel chain? Small/big? Number of rooms? Is it open all year long or just for high season? City or resort hotel?
Data collection: How did you get the 600 questionnaires completed? Did you get customers’ emails from 1 hotel? Did you email them and then got the reply? It not clear for the readers nor researchers wanting to perform a similar research.
RESULTS: Figure 1, quality of the image is difficult to read.
CONCLUSION: Limitations, should be clearly specified that the research has been done using information of 1 hotel in 1 country/location.
REFERENCES, do not follow MDPI guidelines. Abbreviated Journal Names should be used, following ISO 4 standard (e.g. “Front. Public Health” instead of “Frontiers in Public Health” at reference number 9).
Author Response
Response to Reviewer 2 Comments
Manuscript ID: sustainability-1614004
Title: Can natural-based solutions (NBSs) for stress recovery in green hotels affect re-patronage intention?
Dear Editor and reviewers,
Thank you very much for reviewing our manuscript. We appreciate the opportunity you have afforded us to revise and resubmit. We found your suggestions to be thought-provoking and useful and have worked diligently to improve the manuscript as you suggested. Below, you will find our replies and responses to your constructive comments. Within the responses, red sections denote changes we made to the manuscript itself. Within the manuscript itself, changes are highlighted in red. We hope the changes made are satisfactory to you.
******************************************************************************************
Point 1: ABSTRACT: Line 12, satisfaction (SF). Then Line 23, satisfaction (SA). Should it be SA in Line 12?
Response 1: Thank you for the helpful comments. We have corrected it.
Point 2: INTRODUCTION: Satisfaction is briefly mentioned in the Introduction. Line 77 (no abbreviation) and Line 94 (abbreviated). Should this dimension be clarified?
Response 2: Thank you for your comment. We have corrected it and we corrected and supplemented the same mistakes throughout the paper.
Point 3:. CONCEPTUAL FRAMEWORK: some layout and errors, line 213.
Response 3: Thank you for your comment. We have corrected it.
We have corrected it.
Point 4: MATERIALS AND METHODS: Line 270 and 271, November, which year? Line 293, dates are not clear (“November 8 to November 4”?). Lines 294-295, November, year?
Response 4: Thank you for your comment. We have corrected it.
Point 5: It might help for the readers to have some information about the “green hotel”, Seoul Walkerhil Hotel. Is this hotel the subject of the main research? Is it privately owned? part of hotel chain? Small/big? Number of rooms? Is it open all year long or just for high season? City or resort hotel
Response 5: Thank you for your comment, and you raised good points.We have added that the Grand Walkerhill Seoul Hotel is an eco-friendly hotel to the text. Please see line 265-270 in page 6.
Point 6: Data collection: How did you get the 600 questionnaires completed? Did you get customers’ emails from 1 hotel? Did you email them and then got the reply? It not clear for the readers nor researchers wanting to perform a similar research
Response 6: Thank you for your comment. Through an online survey company, we checked whether consumers living in Korea had visited green hotels in the past year, and conducted a survey of consumers who have visited. The survey company that I commissioned is Embrain macromill group (embrain.com). It is doing Korea’s largest number of research projects and is leading the largest panel in Asia (3 million panelists). We have supplemented the text based on the above content. Please see line 292-293 in page 6.
Point 7: RESULTS: Figure 1, quality of the image is difficult to read.
Response 7: Thank you for your comment. We have modified Figure 1 to make it look good. Please see page 10.
Point 8: CONCLUSION: Limitations, should be clearly specified that the research has been done using information of 1 hotel in 1 country/location
Response 8: Thank you for your comment. We did what you commented in the limitations. We have modified and supplemented based on your comments on the limitations. Please see line 469-472 in page 10.
Point 9: REFERENCES, do not follow MDPI guidelines. Abbreviated Journal Names should be used, following ISO 4 standard (e.g. “Front. Public Health” instead of “Frontiers in Public Health” at reference number 9).
Response 9: Thank you for your comment. We have modified and supplemented the reference list in accordance with the ISO 4 standard. Please see in page 13-16.
Round 2
Reviewer 1 Report
The authors addressed the reviewers' comments to my satisfaction. I don't have any further comments on the paper.